# The Neurokinin-1 Receptor Antagonist Aprepitant: An Intelligent Bullet against Cancer?

**DOI:** 10.3390/cancers12092682

**Published:** 2020-09-20

**Authors:** Miguel Muñoz, Rafael Coveñas

**Affiliations:** 1Research Laboratory on Neuropeptides (IBIS), Virgen del Rocío University Hospital, 41013 Sevilla, Spain; mmunoz@cica.es; 2Institute of Neurosciences of Castilla y León (INCYL), Laboratory of Neuroanatomy of the Peptidergic Systems, University of Salamanca, c/ Pintor Fernando Gallego, 1, 37007 Salamanca, Spain

**Keywords:** apoptosis, antitumor, antimetastasis, anti-angiogenesis, NK-1 receptor, substance P, drug repositioning, Emend

## Abstract

**Simple Summary:**

Due to the lack of selectivity and the severe side-effects, cytostatics are the drugs of the present but not of the future and consequently new anticancer strategies must be developed. After binding to the neurokinin-1 receptor (NK-1R), NK-1R antagonists exert antitumor actions (antiproliferative, antimetastasis), are safe and do not cause serious side-effects. Aprepitant, a non-peptide NK-1R antagonist, is currently used in clinical practice as antiemetic but this compound also shows antitumor effects against a broad-spectrum of cancers. Our aim is to review the multiple antitumor actions exerted by aprepitant and to show the potential use of this drug as an antitumor agent. Aprepitant could be considered as an intelligent bullet against cancer. The data support the reprofiling of aprepitant for a new therapeutic use as an antitumor agent. The administration of aprepitant in cancer patients to prevent recurrence/metastasis after surgical procedures, thrombosis and thromboembolism is also suggested.

**Abstract:**

Neurokinin-1 receptor (NK-1R) antagonists exert antitumor action, are safe and do not cause serious side-effects. These antagonists (via the NK-1R) exert multiple actions against cancer: antiproliferative and anti-Warburg effects and apoptotic, anti-angiogenic and antimetastatic effects. These multiple effects have been shown for a broad spectrum of cancers. The drug aprepitant (an NK-1R antagonist) is currently used in clinical practice as an antiemetic. In in vivo and in vitro studies, aprepitant also showed the aforementioned multiple antitumor actions against many types of cancer. A successful combination therapy (aprepitant and radiotherapy) has recently been reported in a patient suffering from lung carcinoma: the tumor mass disappeared and side-effects were not observed. Aprepitant could be considered as an intelligent bullet against cancer. The administration of aprepitant in cancer patients to prevent recurrence and metastasis after surgical procedures, thrombosis and thromboembolism is discussed, as is the possible link, through the substance P (SP)/NK-1R system, between cancer and depression. Our main aim is to review the multiple antitumor actions exerted by aprepitant, and the use of this drug is suggested in cancer patients. Altogether, the data support the reprofiling of aprepitant for a new therapeutic use as an antitumor agent.

## 1. Introduction

Cancer is a major public health problem worldwide, and currently, chemotherapy is the basis of the pharmacological treatment against it, despite the display of severe and numerous side-effects and the possible effects on the whole body, including those on essential organs (e.g., the lungs, brain and heart). In addition, the overall contribution of chemotherapy to five-year survival has been reported as between 2.1% and 2.3% [1]. Thus, it is crucial to investigate new targets and specific antitumor therapeutic strategies directed exclusively against cancer cells to avoid serious side-effects in cancer-suffering patients. Currently, there are many data demonstrating the involvement of the substance P (SP)/neurokinin-1 receptor (NK-1R) system in cancer [2,3]; one of these targets could be the NK-1R, and thus, a new antitumor strategy could be the use of NK-1R antagonists (e.g., the drug aprepitant), since it is known that SP (after binding to the NK-1R) promotes mitogenesis in tumor cells and that NK-1R antagonists (after binding to the same receptor) counteract the SP-mediated mitogenesis and induce apoptotic mechanisms in these cells [2,3].

Non-peptide and peptide NK-1R antagonists have been reported. Peptide antagonists (e.g., NY-3460, NY-3238, Spantide I and II) show poor potency, neurotoxicity, a poor ability to discriminate between tachykinin receptors, an inability to cross the blood–brain barrier and metabolic instability, and for these reasons, the use of them in clinical practice is very limited [2,3]. Non-peptide NK-1R antagonists (e.g., CP-96,345, CP-99,994, L-732,138, WIN-51,708, L-733,060, L-742,694 and aprepitant) show a high affinity for the NK-1R and cross the blood–brain barrier, and hence, these compounds have been used for the treatment of a broad range of disorders [2,3]. NK-1R antagonists exert anxiolytic, anti-inflammatory, antidepressive, analgesic, antiviral and antitumor actions [2,3].

Aprepitant (5-[[(2*R*, 3*S*)-2-[(1*R*)-1-[3,5-bis(trifluoromethyl) phenyl] ethoxy]-3-(4-fluorophenyl) -4-morpholinyl]methyl]-1,2-dihydro-3 *H*-1,2,4-triazol-3-one, MW 534.43; Emend, MK-869, L-754,030; oral administration; approved by the U.S. Food and Drug Administration in 2003) is a highly selective non-peptide NK-1R antagonist that binds to the human NK-1R (also known as the SP receptor). In radio-ligand binding assays, aprepitant was approximately 3000-fold more selective for the human cloned NK-1R (IC_50_ = 0.1 nM) versus the human cloned NK-3R (IC_50_ = 300 nM) and 45,000-fold versus the human cloned NK-2R (IC_50_ = 4500 nM) [4]. A morpholine nucleus introduced in the NK-1R antagonist L-742,694 (a benzylether piperidine) enhanced the receptor-binding affinity, and to prevent metabolic deactivation, chemical changes (e.g., fluorination of the phenyl ring and methylation on the phenyl ring) were performed. These changes led to the compound aprepitant. This drug exerts antiemetic, antipruritic, antiviral and antitumor effects. In fact, in in vitro and in vivo experiments, it has been demonstrated that aprepitant exerts an antitumor action by inducing apoptosis in tumor cells. This has been observed in in vitro experiments in many human cancer cell lines (T-ALL BE-13 and B-ALL SD-1 (acute lymphoblastic leukemia); MT-3, MCF-7, MDA-MB-468 and BT-474 (breast cancer); SW-403 (colon carcinoma); 23132/87 (gastric carcinoma); GAMG (glioblastoma); HepT1, H4H6 and HepG2 (hepatoblastoma); HEp-2 (laryngeal carcinoma); H-69 and COR-L2 (lung cancer); MEL HO, COLO 858 and COLO 679 (melanoma); SKN-BE, IMR-32 and KELLY (neuroblastoma); MG-63 (osteosarcoma); PA-TU-8902 and CAPAN-1 (pancreatic carcinoma); Y-79, WERI-Rb-1 (retinoblastoma)) [2,3]. Moreover, the antitumor action of aprepitant/fosaprepitant has been confirmed (both compounds decreased the volume of the tumor) in experimental mouse models (MG-63 human-osteosarcoma and HuH6 human-hepatoblastoma xenografts). Finally, several studies have been reported on the use of aprepitant (in combination with radiotherapy) in a patient suffering from lung cancer [5] or in nuclear medicine for the diagnostic/treatment of tumors expressing the NK-1R [6,7]. Both therapeutic strategies have shown promising results. In this line, a recent review has focused on the biological and chemical aspects of peptide and non-peptide NK-1R ligands (e.g., aprepitant) as radiopharmaceuticals and on the application of these compounds in targeted radionuclide cancer therapy [6]. Moreover, another recent study has been focused on the synthesis and evaluation of new radioconjugates of aprepitant [7]. Both studies have shown the potential use of aprepitant in nuclear medicine for the diagnosis/treatment of NK-1R-positive tumors. As aprepitant is a poorly water-soluble drug, many strategies have been developed to enhance its solubility and dissolution (e.g., using a phosphatidylcholine-based solid dispersion system) [8]. To increase exposure and to minimize food effects, aprepitant has been developed as a nanoparticle formulation, which increased its bioavailability three-/four-fold [9]. Recently, a high-resolution crystal structure of the human NK-1R bound to aprepitant has been published [10]; this information will serve well in the future for the design of new drugs that target the NK-1R with a higher activity, selectivity and affinity. Cinvanti (an aprepitant injectable emulsion) has been also approved for clinical use, whereas fosaprepitant dimeglumine (Ivemend) is a water-soluble prodrug of aprepitant that is administered intravenously and converted into aprepitant by ubiquitous phosphatases. Acting at the NK-1Rs of the neurons located in the area postrema/nucleus of the tractus solitary and gastrointestinal tract, aprepitant blocks the vomiting reflex. In clinical practice, aprepitant is used for the prevention of both chemotherapy-induced and post-operative nausea and vomiting. In the first case, aprepitant is generally combined with a glucocorticoid (e.g., dexamethasone) and a 5-hydroxytryptamine type-3 (5-HT_3_) receptor antagonist. Aprepitant is administered once before chemotherapy (Day 1: 125 mg; Days 2 and 3: 80 mg), whereas fosaprepitant (150 mg) is only administered once 30 min before chemotherapy. For the treatment of chemotherapy-induced nausea and vomiting in pediatric patients, the use of either aprepitant (one hour before chemotherapy; Day 1: 3 mg/kg; Day 2: 2 mg/kg) or fosaprepitant has also been approved [11,12,13]. For the prevention of anesthesia-induced post-operative nausea/vomiting, aprepitant (40 mg) is used but fosaprepitant is not currently approved, and in pediatric patients, neither aprepitant nor fosaprepitant has been approved for the treatment of postoperative nausea and vomiting. The half-life of aprepitant ranges from 9 to 13 h, and it is metabolized in the liver by cytochrome P450, family 3, subfamily A (CYP3A4).

Cancer cells express the mRNA for the NK-1R [14,15,16,17], and cancer-suffering patients show an up-regulation of the NK-1R mRNA expression when compared to healthy subjects [18]. Consequently, an important question arises: could aprepitant be used for the treatment of other pathologies in which the NK-1R is up-regulated? This question is an important one, as considerable data exist showing that NK-1R antagonists (e.g., aprepitant and L-733,060) could be used for the treatment of virus infections, pruritus and cancer [19,20,21,22]. It is also known that aprepitant suppresses both coughing in lung-cancer patients [23,24] and the inflammatory response related to rheumatoid arthritis [25] and that, by promoting an anti-inflammatory effect, it also protects endothelial cells from the inflammatory response and injury [26]. To date, it is known that cancer cells (independently of the tumor type) show a common phenomenon: the overexpression of the NK-1R [2,3,19]. This is extremely important since it means that the NK-1R is a common target for the treatment of any tumor type. In addition, many data have demonstrated the involvement of the SP/NK-1R system in cancer [2,3,19,27,28]. For example, SP is a universal mitogen in tumor cells, and in a concentration-dependent manner, non-peptide NK-1R antagonists (e.g., aprepitant, L-733,060, L-732,138, CP 96,345 and SR-140,333; it is important to note that these compounds show different chemical compositions) exert antitumor actions against many human cancer cell lines (e.g., glioma, neuroblastoma, osteosarcoma, retinoblastoma, breast cancer (including triple-negative breast cancer cells), gastric cancer, pancreatic cancer, lung cancer (small- and non-small-cell lung cancer), colon cancer, larynx cancer, B- and T-cell acute lymphoblastic leukemias and acute myeloid leukemia) [2,3,15,16,17,19,28]. In general, it has been demonstrated that the antitumor action mediated by L-733,060 is more potent than that exerted by aprepitant and that the antitumor action of aprepitant is more potent than that of L-732,138 [29]. It has been also shown (in vitro and in vivo) that aprepitant/fosaprepitant increased tumor cell death and decreased tumor cell viability/cell proliferation and tumor volume [16,30] and that NK-1R antagonists induced the death of chemoresistant cancer cells [31].

Accordingly, the main goal of this review is to show the broad-spectrum antitumor action that characterizes the approved antiemetic drug aprepitant and to suggest its use as an antitumor drug. In other words, the reprofiling of this drug as a universal antitumor agent is suggested. In oncology, drug repositioning is an excellent and rapid alternative therapeutic strategy for new treatments against cancer, and in addition, the risk for patients is greatly reduced [32]. A recent review collected many studies performed to know whether non-antitumor drugs used in clinical practice could also exert anticancer effects [32]. In fact, the repositioning of aprepitant and eight other drugs approved for non-oncological indications (e.g., captopril and ritonavir) has been carried out for the treatment of recurrent glioblastoma (CUSP9 strategy: Wnt activity is decreased) [32,33]. For the repositioning of aprepitant as an antitumor drug, the dose and days of administration must be increased compared to those for the standard antiemetic treatment currently used in clinical practice. This will be also mentioned and discussed.

## 2. Cancer and the Substance P/Neurokinin-1 Receptor System

SP is an undecapeptide that, as hemokinin-1 (HK-1), binds to the NK-1R (a G-protein-coupled receptor containing seven hydrophobic alpha-helical transmembrane domains). Both peptides, members of the tachykinin family of peptides, are the natural ligands of the NK-1R and promote the proliferation and migration of tumor cells. In these cells and via the protein kinase A or C pathway, HK-1 promotes the phosphorylation of Akt, jun N-terminal kinase (JNK), p38 and extracellular signal-related kinases (ERK) 1/2, leading to the activation of nuclear factor kappa B (NF-κB) and activator protein 1 (AP-1) [34,35], and increases the levels of matrix metalloproteinases 2 and 14 and membrane type 1-matrix metalloproteinase, facilitating the migration of tumor cells [34,35,36]. SP and the NK-1R are widely distributed in the body (e.g., in the central and peripheral nervous systems and immune cells), and the peptide has also been found in breast milk, cerebrospinal fluid and blood.

Cancer cells express both SP and the NK-1R, and it is known that the SP/NK-1R system plays an important role in cancer since it is involved in tumor growth and development (both solid and non-solid) [2,28]; this has been demonstrated in many in vitro and in vivo experiments [2,3,16,17,27,28]. SP, via the NK-1R, exerts, in cancer cells, the following actions: mitogenesis, migration (invasion and metastasis), anti-apoptotic effects and an increase in the glycolytic rate (tumor cells increase their metabolism due to the glucose obtained; this mechanism is called the Warburg effect). In other words, the SP signal promotes beneficial effects for the survival of tumor cells. By an autocrine mechanism, it has been demonstrated that the SP released from cancer cells induced the growth of tumors [37] and that this autocrine action promotes persistent human epidermal growth factor receptor 2 (HER2) activation, leading to drug resistance and malignant progression [38]. SP favors the proliferation of cancer cells via the mitogen-activated protein kinase (MAPK) pathway and the transcriptional inhibitor of the Notch signaling pathway named hairy and enhancer of split 1 (Hes 1) [39,40]. The latter is important since Hes 1 decreased the growth suppression of cancer cells when a downregulation of the NK-1R occurred [40]. SP promotes the mitogenesis and migration of tumor cells by phosphorylating and activating the ERK1/2 [41]. It has also been demonstrated that β-arrestin is essential for the NK-1R-mediated proliferation of cancer cells and for the G2/M phase transition and that a deficiency in β-arrestin augmented the sensitivity of tumor cells to NK-1R antagonists [42]. In addition, in endothelial cells expressing the NK-1R, SP promotes the proliferation of these cells and thus favors angiogenesis, and by increasing the tumoral blood supply, the development of the tumor is also favored. SP also regulates transcription factors and proto-oncogenes (e.g., AP-1, c-myc and c-fos) involved in cellular transformation and differentiation, apoptosis and cell-cycle progression, and the undecapeptide also promotes the synthesis of pro-inflammatory cytokines [39,43]. These cytokines, released into the microenvironment of the tumor, amplify the inflammatory response mediated by SP [18,44,45]. This is important because increased tumor progression has been related to the levels of cytokines [46,47,48] and neutral endopeptidases that cleave SP, blocking the invasive capacity of cancer cells by decreasing the tumor microenvironmental level of SP [49]. In summary, SP promotes the growth and development of tumors and favors angiogenesis, since both cancer and endothelial cells express the NK-1R. However, it is important to note that the NK-1R is overexpressed in any cancer cell type, and thus, the receptor is a crucial target for cancer treatment [2,3,18,27]. This is essential for the establishment of a common antitumor strategy irrespective of cancer type: the use of NK-1R antagonists (e.g., aprepitant) as broad-spectrum antitumor drugs. In addition, the overexpression of the NK-1R in cancer cells could be used as a potential tumor biomarker that could aid rapid disease diagnosis/treatment. Current data suggest that NK-1R antagonists could exert a double therapeutic effect: antitumor and anti-angiogenic.

SP, via the NK-1R, regulates another important action, the migration of tumor cells, which is mediated by the protein kinase B (Akt)/NF-κB pathway [50,51,52]. SP controls membrane blebbing (important for cell migration/spreading) and the synthesis of degradative enzymes such as matrix metalloproteinases; both mechanisms facilitate metastasis [53,54,55]. SP promotes cell invasion by increasing the expression of matrix metalloproteinases (2 and 9), but this can be blocked with NK-1R antagonists [53,55]. SP accelerates the migration of esophageal squamous cell carcinoma (in which the NK-1R is overexpressed) and enhances the expression of factors involved in angiogenesis via the overexpression of vascular endothelial growth factor receptor 1 and vascular endothelial growth factor A [56]. It has been reported that by reducing the activation of Akt or NF-κB, the migration of cancer cells was attenuated [50]. It is also known that the expression of the full-length isoform of the NK-1R reduced metastasis, whereas the activation of the truncated form increased it [57]; that SP promoted the migration of cancer cells by modulating the level of intracellular Ca^++^ [58]; and that in metastatic tumors, an increase in SP/NK-1R staining occurred [30].

Despite the numerous findings demonstrating the involvement of SP in the proliferation/migration of tumor cells, some results showed an opposite effect. Thus, it has been reported that SP neither altered the proliferation of tumor cells nor reverted the antitumor effect of NK-1R antagonists and that SP exerted an antimetastatic effect [59,60]. Moreover, there are data suggesting that SP increases the antitumor immune response in a murine model of breast cancer metastasis and that the activation of the sensory fibers containing SP may block metastasis [61,62]. In addition, another study showed that aprepitant did not exert a cytotoxic effect against cancer cells, although the authors indicated that this finding could be due to the experimental model (brain tumor secondary to breast cancer) and tumor cell lines used [63]. These discrepancies could be due to methodological procedures and must be clarified in future studies.

As indicated above, two isoforms (full-length and truncated) of the NK-1R have been reported, and in less-differentiated and more advanced tumors, the expression of the NK-1R is higher [64]. In tumor cells, the expression of the truncated form is higher than that of the full-length form [18], and it has been suggested that the malignant phenotype of tumor cells is due to a low expression of the full-length form [65]. In fact, the truncated isoform has been related to malignant transformation in colitis-associated cancer [66], and the overexpression of the truncated isoform promoted the malignant transformation of non-tumorigenic cells [57]. The expression of the full-length isoform has been inversely related to the invasiveness, metastasis and proliferation of tumor cells [57], and its downregulation blocked these mechanisms [67]. The up-regulation of the truncated form is important because it is involved in the growth and malignancy of tumor cells and in the synthesis of cytokines with growth-promoting actions, whereas the full-length form promotes a slow growth of tumor cells [68,69,70]. In tumors, a high level of both cytokines and NF-κB has been observed, and it is known that NF-κB slightly increases the full-length form but, on the contrary, up-regulates the truncated receptor. It is important to note that in cancer cells, the activation of the full-length isoform blocked the proliferation of these cells, whereas the activation of the truncated form induced its proliferation [57]. In cancer cells, tumor growth factor β and the truncated form promoted the proliferation of these cells and inhibited apoptotic mechanisms [71]. Moreover, it was demonstrated that tumor growth factor β regulates, via the protein Smad4 (Smad family member 4), the expression of the truncated form and that aprepitant attenuated the effects of tumor growth factor β on tumor cell proliferation [71]. In addition, two important findings must be highlighted in cancer cells: the expression of the NK-1R is crucial for the viability of tumor cells (after its blockade, tumor cells die by apoptosis) and SP increases the expression of the NK-1R but not that of other tachykinin receptors (e.g., NK-2R) [14,15,17,72]. It is also known that in the case that tumor cells do not receive the stimulus mediated by SP (e.g., when using anti-SP antibodies), the number of apoptotic cells increases; the MAPK signaling pathway, the steady state of HER2 and the endothelial growth factor receptor decrease; and the synthesis of cell-cycle-progression/cell-cycle-related proteins (e.g., mammalian target of rapamycin (mTOR)) is inhibited [31,38]. The data show the crucial importance of the SP stimulus for tumor cells.

According to the aforementioned, an important question arises: could the SP/NK-1R system be used as a predictive factor in cancer? To date, it has not been fully demonstrated but it has been suggested in emerging data (Table 1). In cancer-suffering patients, it has been reported that the level of SP in serum was higher than that in healthy subjects, and a higher number of NK-1Rs has been associated with cancer stage, larger tumor size, tumor-node metastasis, higher invasion/metastatic potential and poor prognosis [40,64,65,73,74]. In breast cancer, the high expression of the truncated isoform of the NK-1R has been correlated with poor prognosis [71]. Moreover, it has been reported that the expression of the full-length NK-1R was inversely correlated with lymph-node metastasis and tumor-lymph-node metastasis [65] and that SP/NK-1R expression may be used as a predictor for the prognosis of colorectal cancer [75]. The number of fibers containing SP has been correlated with lymph-node metastasis, tumor size and cancer differentiation [58,76], and it is known that the SP expression increased as the proliferation of cancer cells increased [77]. Another important finding is that the number of fibers containing SP in the transitional zone between normal and cancer-invaded areas decreased, probably due to the release of SP from these fibers [78]. In oral precancerous epithelium, SP nuclear and cytoplasmic expression in non-tumor epithelium has been associated with the presence of epithelial dysplasia and carcinoma [79], and the hypermethylation of the tachykinin receptor type 1 and tachykinin-1 has been suggested as a biomarker for neck and head cancer [80]. Finally, in malignant tumors, the expression of SP and the NK-1R has positively been correlated with a higher expression of Ki-67, and the latter has positively been correlated with the rate of growth of an odontogenic tumor [81]. Overall, the data suggest that the SP/NK-1R system plays an important role as a predictive factor for cancer (Table 1).

## 3. Aprepitant: Antitumor Action and Signaling Pathways

Aprepitant, in a concentration-dependent manner, is a broad-spectrum antitumor drug, as it promotes apoptotic mechanisms (Table 2) in many human cancer types such as colon, gastric, laryngeal and pancreatic carcinomas; esophageal squamous cell carcinoma; B- and T-cell acute lymphoblastic leukemias; retinoblastoma; neuroblastoma; glioma; osteosarcoma; acute or chronic myeloid leukemias; melanoma; breast cancer; and hepatoblastoma and lung cancer (small and non-small cell) [14,15,16,17,27,28,56,72,82,83,84,85]. In many cases, maximum inhibition (100%) was observed when aprepitant was administered at a concentration of ≥70 µM (Table 2) [83]. It is important to note that as aprepitant is lipid soluble and crosses the blood–brain barrier, this NK-1R antagonist could also be used for the treatment of tumors located inside the central nervous system. In this sense, a synergy has been reported for glioma treatment between ritonavir (an antiviral drug) and aprepitant (Table 2) [86]. Both compounds interfere with Akt signaling, and this could help to explain the exertion of strong synergy (Table 2).

Which antitumor actions are mediated by aprepitant? In cancer cells, aprepitant promotes G2/M-phase cell-cycle arrest [87] and apoptosis by favoring Ca^2+^ flux from the endoplasmic reticulum into mitochondria and increases the level of mitochondrial reactive oxygen species, leading to functional impairment (Table 2) [18]. In fact, the viability of tumor cells (treated with NK-1R antagonists) increased when reactive oxygen species scavengers were administered, whereas SP did not favor the production of these species and induced a weak mitochondrial Ca^++^ flux [18]. That is, the Ca^++^ introduced into the mitochondria is an apoptotic stimulus [88]. In tumor cells, aprepitant decreases p70 S6 kinase phosphorylation and attenuates the activation of the mTOR signaling axis (Table 2) [89], whereas SP activates mTOR and increases tumor cell growth and metastasis by activating p70 S6 kinase and the eukaryotic initiation factor 4E-binding protein 1 (4E-BP1) [90]. In cancer cells, aprepitant increases the sensitization of these cells to the cytotoxic action of determined substances (e.g., arsenic trioxide and vincristine); impairs the interaction of Forkhead box M1 with beta-catenin, leading to the blockade of the Wnt canonical pathway; arrests the G2 cell cycle (promoting apoptosis); activates the caspase 3-dependent apoptotic pathway; downregulates the expression of cyclin D1 and lymphoid enhancer-binding factor 1; and alters the DNA replication rate and cell cycle (Table 2) [89,90,91,92,93,94]. Fosaprepitant also sensitizes neuroblastoma cells to cytotoxic agents (e.g., etoposide and doxorubicin) and exerts a synergic antitumor action [95]. In cancer cells, aprepitant promotes apoptotic mechanisms since it increases the cleaved poly (ADP-ribose) polymerase (88 kDa) and caspase 3 (18 kDa) forms (Table 2) [17].

Moreover, aprepitant increases the mRNA expression of p21, p73, Bax, Bid and Bad (pro-apoptotic targets), increases the expression of apoptotic markers (e.g., propidium iodide and annexin-V), and decreases the expression of c-myc (the molecule by which p73 regulates NF-κB activity; aprepitant shows a diminished antitumor action when the NF-κB pathway is overactivated) (Table 2) [94]. This is significant, as one of the mechanisms involved in the resistance of cells to aprepitant is currently known and because of the increased expression of the NK-1R due to the activation of NF-κB by SP [69]. In esophageal squamous cell carcinoma, aprepitant arrests cells in the G2/M phase, inhibits the Akt/phosphatidylinositol 3-kinase (PI3K) axis (including its downstream effectors, e.g., NF-κB) and induces apoptosis (Table 2) [96]. In breast cancer cells, aprepitant increases the activity of caspases 3, 8 and 9, indicating that the drug promotes apoptotic mechanisms (Table 2) [87], whereas fosaprepitant administered to mice xenografted with myeloid leukemia cells increases the median survival from four to seven days [82]. Aprepitant also reduces the tumor burden in neuroblastoma xenografted tumors [97], decreases the death of cardiomyocytes induced by doxorubicin and increases tumor cell sensibility to doxorubicin (Table 2) [98], and in the same way, fosaprepitant reduces the volume of tumors (e.g., neuroblastoma and osteosarcoma) [16,95]. In hepatoblastoma, the Wnt and Akt signaling pathways are important targets of aprepitant, as this NK-1R antagonist blocks the canonical Wnt pathway (a downregulation of the leucine-rich repeat-containing G-protein coupled receptor 5 (LGR5) and axis inhibition protein 2 (AXIN2) Wnt target genes occurs along with a decrease in β-catenin) and reduces the phosphorylation of p70S6K/4E-BP1/2 (Table 2) [89]. The result of these actions is that the growth of the hepatoblastoma cells is decreased. Moreover, aprepitant downregulates the expression of the FOXM1protein, which is involved in the translocation of β-catenin into the nucleus, and this promotes apoptotic mechanisms and growth arrest (Table 2) [89,99,100]. In an experimental model (xenografted hepatoblastoma), the administration of aprepitant (80 mg/kg/day for 24 days) decreased the weight and tumor volume, the vascularized area, the serum level of tumor-specific alpha-fetoprotein (a marker of hepatoblastoma) and the number of cells expressing Ki-67 (Table 2) [85]. In colon cancer, aprepitant also blocked the canonical Wnt pathway by decreasing super TOP/FOP and by increasing the membrane stabilization of β-catenin [91]. This was independent of the β-catenin mutational status and of Wnt baseline activity. Aprepitant reduced the expression of liver stemness markers (octamer-binding transcription factor 4 (OCT4), NANOG (a homeobox protein), sex determining region Y-box 2 (SOX2), cluster of differentiation 13 (CD13) and alpha-1-fetoprotein (AFP)) (Table 2) [89]; this is important since cancer stem cells are involved in cancer resistance/relapse [101], and the treatment (with aprepitant) of colon cancer cells grown under cancer stem cell conditions decreased both the size and number of spheres formed [91]. SP is involved in peritumoral edema and in early blood–brain barrier disruption (SP induces changes in the localization and distribution of tight junctions in the brain microvascular endothelial cells), whereas aprepitant decreased brain water content and blood–brain barrier permeability (Table 2) [92,102], and in brain tumors, the same drug increased cell death and reduced cell proliferation and tumor volume [30]. Thus, it seems that the invasion of tumor cells into the brain is an SP-mediated mechanism [103]. Aprepitant exerts an antitumor activity against acute/chronic myeloid leukemia cells; the drug decreases colony forming and promotes apoptotic mechanisms [104], and it has thus been suggested that the use of aprepitant (in combination with chemotherapeutic drugs or alone) could also serve as an excellent therapeutic strategy for the treatment of leukemia [94]. Aprepitant also exerts an antinociceptive effect in myeloid leukemia-induced bone pain since the NK-1R antagonist decreases inflammatory processes (Table 2) [18], and the drug, by blocking the NK-1Rs expressed in macrophages, inhibits inflammatory mechanisms [105]. Aprepitant prevents the SP-dependent phosphorylation of ERK1/2 and Akt [65], blocks the Akt/p53 pathway [106] and inhibits tumor cell migration and angiogenesis by blocking the overexpression of matrix metalloproteinases 2 and 9, vascular endothelial growth factor receptor 1 and vascular endothelial growth factor A by tumor cells (Table 2) [56]. Finally, it has been demonstrated that aprepitant increases the phosphorylation of the epidermal growth factor receptor and that the drug induces cytotoxicity in human keratinocytes, contributing to its antitumor effects [20].

**Table 2 cancers-12-02682-t002:** Effects of aprepitant on cancer cells.

Promotes apoptosis: inhibition (100%) observed after administration of aprepitant (≥ 70 µM) [83]
Promotes apoptosis: favors Ca^2+^ flux from the endoplasmic reticulum into mitochondria [18]
Exerts an antitumor synergic effect with ritonavir [86]
Exerts an antinociceptive effect in myeloid leukemia-induced bone pain [18]
Increases the level of mitochondrial reactive oxygen species [18]
Increases the sensitization of tumor cells to arsenic trioxide or vincristine [94]
Increases the cleavage of poly (ADP-ribose) polymerase [17]
Increases the levels of mRNA expression of p21, p73, Bax, Bid and Bad (pro-apoptotic targets) [94]
Increases the expression of apoptotic markers (propidium iodide and annexin-V) [94]
Activates the caspase 3-dependent apoptotic pathway and increases the activity of caspases 8 and 9 [17,87,93]
Alters DNA replication rate and cell cycle. Promotes G2/M-phase cell-cycle arrest [87,90,96]
Prevents the SP-dependent phosphorylation of ERK1/2 and Akt [65,106]
Interferes with the Akt/p53/PI3K signaling pathway [96]
Attenuates the activation of mammalian target of rapamycin signaling axis [89]
Blocks the canonical Wnt pathway (downregulation of Wnt target genes LGR5 and AXIN2 occurs along with a decrease in β-catenin) [89]
Impairs the interaction of Forkhead box M1 with beta-catenin, blocking the Wnt canonical pathway [90]
Inhibits tumor cell migration and angiogenesis by blocking the overexpression of matrix metalloproteinases 2 and 9, vascular endothelial growth factor receptor 1 and vascular endothelial growth factor A [56]
Decreases p70S6 kinase/4E-BP1/2 kinase phosphorylation [89,90]
Decreases the expression of c-myc [94]
Decreases the death of cardiomyocytes induced by doxorubicin and increases tumor cell sensibility to doxorubicin [98]
Decreases weight/tumor volume, vascularized area, serum level of tumor-specific alpha-fetoprotein and the number of cells expressing Ki-67 [85]
Decreases brain water content and blood–brain barrier permeability [91,102]
Shows a diminished antitumor action when NF-κB pathway is overactivated [94]
Reduces the expression of lever stemness markers (OCT4, NANOG, SOX2, CD13 and AFP) [89]
Downregulates the expression of cyclin D1 and lymphoid enhancer-binding factor 1 [90]
Downregulates the expression of the FOXM1 protein involved in the translocation of β-catenin into the nucleus [89,99,100]
Counteracts glycogen breakdown, that is, the Warburg effect [3]

## 4. Relationships between Aprepitant and Other Drugs

It is important to note that the co-administration of chemotherapeutic drugs and aprepitant exerts a more effective antitumor action than the administration of aprepitant alone [94]. Anthracyclines are used to treat tumors, but for example, treatment with doxorubicin induces cardiotoxicity [107]. Doxorubicin cardiotoxicity is mediated by the SP/NK-1R system, and it is known that aprepitant decreases cardiotoxicity and increases the sensitivity of tumor cells to doxorubicin (Table 2) [98]. Moreover, in preclinical studies, it has been demonstrated that aprepitant exerts a protective role against the hepatotoxicity and nephrotoxicity induced by the chemotherapeutic drug cisplatin [108] and that aprepitant (2 mg/kg/day for 12 weeks) inhibited the cutaneous (e.g., nose crusting, scabbing, skin reddening and alopecia) and neurogenic inflammation side-effects mediated by erlotinib (an epidermal growth factor receptor-tyrosine kinase inhibitor used as an anticancer treatment) [109]. Erlotinib increases the level of SP, which mediates the side-effects observed, whereas aprepitant mitigates it, including causing a decrease in the number of NK-1Rs expressed in the skin. It is also known that inhibitors of the epidermal growth factor receptor also promoted cutaneous side-effects (e.g., pruritus) and that aprepitant exerted an antipruritic effect by activating this receptor [20].

It is known that one of the adverse effects induced by treatment with vincristine (a CYP3A4 substrate used in chemotherapy) is the appearance of peripheral neuropathy [110]. Aprepitant is a moderate CYP3A4 inhibitor; this inhibition causes, during the therapeutic treatment, an increase in the concentration of vincristine, inducing toxicity. The study shows that the co-administration of aprepitant or fosaprepitant and vincristine increases the risk of chemotherapy-induced peripheral neuropathy [110]. As aprepitant is an inhibitor (dose-dependent) and inducer of the cytochrome P450 CYP34A family of enzymes, substances metabolized by this family can be affected. This is the case for dexamethasone, oral hormonal contraceptives, benzodiazepines, ketoconazole and warfarin [111], and thus, a relative contraindication occurs (Table 2). Aprepitant may decrease the efficacy of hormonal contraceptives, but it did not alter the pharmacokinetic of cyclophosphamide [111]. In patients suffering from multiple myeloma and treated with chemotherapy, aprepitant did not alter the pharmacokinetics of a high dose of melphalan [112]. Moreover, this NK-1R antagonist did not alter the metabolism of vinorelbine, palonosetron, thiotepa, ondansentron, digoxin, hydrodolasetron, dinaciclib or granisetron, but the doses of methylprednisolone or dexamethasone must be decreased when co-administered with aprepitant (corticosteroid plasma levels are increased by the drug) [111]. It is known that the plasma levels of some chemotherapeutic agents (e.g., paclitaxel, docetaxel, midazolam and irinotecan) are increased by aprepitant [111]. Finally, in cancer cells but not in normal cells, a synergic effect between microtubule-destabilizing agents (e.g., vinblastine) and NK-1R antagonists has been reported [113,114].

## 5. Safety

Both aprepitant and fosaprepitant are well tolerated and safe, and the use of both non-peptide NK-1R antagonists shows few contraindications (e.g., the co-administration of cisapride/pimozide, hypersensitivity) (Table 3) [13,111,115]. The therapeutic index of aprepitant is very high, and cases showing a toxic effect due to an overdose of the drug are rare (Table 3). In healthy subjects, neither fosaprepitant nor aprepitant affected QTc intervals (Table 3) [111]. The side-effects of aprepitant/fosaprepitant are minimal, and the most common (incidence higher than 10%) are constipation, fatigue, headache, anorexia, hiccups and diarrhea. Less common side-effects (incidence: 1–10%) are hypotension, fever, dehydration, pharyngolaryngeal pain, mucosal inflammation, hot flashes, anemia, dyspepsia, bradycardia, stomatitis, dizziness, neutropenia and insomnia, whereas rare side-effects (incidence >1%) are euphoria, vivid dreams, malaise, lethargy, febrile neutropenia, candidiasis, weight gain, acid reflux, chills, epigastric discomfort, staphylococcal infection, disorientation, polydipsia, cognitive disorders, somnolence and conjunctivitis [111,116].

In several experiments, the safety of aprepitant has been demonstrated. Thus, the proliferation of normal cells (e.g., lymphocytes) was not affected after the administration of a high dose of aprepitant [28,82], and in patients, the administration of high doses of aprepitant (300 mg/day; 1140 mg/day for 45 days) was safe and well tolerated (Table 3) [5,111,117]. It has also been reported that the IC_50_ for lymphocytes is ten-fold higher than that for cancer cells, and consequently, the damage exerted by aprepitant was higher in cancer cells than in normal cells (Table 3) [82]. The safety of aprepitant for human fibroblasts has been also reported; thus, the IC_50_ for fibroblasts was higher than the IC_50_ for cancer cells [83], and it has been also shown that the IC_50_ of aprepitant for human breast epithelial cell lines was >90 μM, three times higher than the IC_50_ for breast cancer cells and even higher than the IC_100_ for these tumor cells (Table 3) [17]. In general, for non-tumor cells, the IC_50_ and IC_100_ of aprepitant are 60 and 90 μM, respectively [83], whereas in general, the IC_100_ of aprepitant for cancer cells is 50 µM (Table 3) [3]. Currently, the molecular mechanisms involved in the resistance to aprepitant of normal cells are unknown. However, the total number of NK-1Rs expressed as well as the rate of full-length/truncated isoforms expressed could explain this resistance. It has been shown that NK-1R antagonists are very effective against tumor cells expressing a higher level of the truncated isoform of the NK-1R [59], and it is known that normal cells express a lower number of the truncated form than cancer cells and that, in the latter cells, the expression of the truncated form is higher than that of the full-length [18,82]. This must be elucidated in future work, as it is a crucial point for understanding the antitumor effects mediated by aprepitant in greater depth. Moreover, a case report has been published in which a patient, suffering from brain metastasis with nausea and vomiting and refractory to all standard antiemetic therapy, received 80 mg/day of aprepitant (initially, for seven months), followed by an increase to 120 mg every third day [118]. An improvement in clinical condition and in the level of the CA153 tumor marker (decreasing from 187 to 122 U/ml) was observed, including good control of nausea/vomiting and no reported side-effects [118]. A clinical trial [119] has shown that aprepitant was safe in cancer patients receiving doxorubicin and ifosfamide chemotherapy and that in patients suffering from depression, the tolerability and safety of aprepitant (300 mg/day) were similar to placebo [117]. It is also important to note that, in cancer cells, a synergism was reported when an NK-1R antagonist was co-administered with cytostatic drugs (e.g., cisplatin, ifosfamide, mitomycin, adriamycin and doxorubicin); however, this effect was not observed in human fibroblasts, and when the latter cells were treated with an NK-1R antagonist, prior to exposure to cytostatic drugs, human fibroblasts were partially protected from these drugs [16]. The same result was observed when non-tumor HEK-293 cells were treated, before exposure to cytostatics, with an NK-1R antagonist [16].

As the cytostatic drugs (e.g., cyclophosphamide, doxorubicin and cisplatin) used in chemotherapy are not specific against cancer cells, these drugs show a very low safety profile, limiting their clinical use and, in addition, promoting serious side-effects. Chemotherapy promotes the release of SP, favoring nausea/vomiting—an effect that is inhibited by NK-1R antagonists (e.g., aprepitant), which exert antiemetic action, and it is also known that radiotherapy increases the level of SP and that treatment with endostatin (an angiogenesis inhibitor) decreases the level of SP; additionally, the inhibitor blocks the growth of tumor cells, and this is probably due to the reduced level of SP [120]. Chemotherapy and radiotherapy induce neurogenic inflammation promoted by the release of SP from nerve terminals. In fact, via the SP/NK-1R system, cyclophosphamide or radiotherapy induces, respectively, inflammatory mechanisms in the gastrointestinal tract or urinary bladder [121]. However, NK-1R antagonists decreased the inflammation induced by cyclophosphamide, and when these antagonists were administered in addition to chemotherapy or radiotherapy, a decrease in the side-effects mediated by chemotherapy/radiotherapy was observed and a synergic antitumor effect was also reported [113,121]. The anti-inflammatory effect mediated by NK-1R antagonists is important, as it is known that both chemotherapy and radiotherapy promote mucositis and mucosal barrier breakdown (a gateway for germs), inducing a systemic infection that is exacerbated by neutropenia. Cisplatin, another chemotherapeutic drug widely used for the treatment of cancer, induces a serious side-effect (nephrotoxicity), in addition to vomiting, bone marrow suppression and hair loss. In experimental animals, it has been demonstrated that NK-1R antagonists (e.g., aprepitant) counteracted the nephrotoxicity and hepatotoxicity mediated by cisplatin [108,122]. Thus, the treatment with cisplatin increased oxidative stress and the levels of tissue cytokines and serum enzymes and induced inflammatory infiltration and necrosis in both the liver and kidney, whereas aprepitant normalized all the parameters studied (e.g., kidney and liver oxidative parameters and inflammatory cytokines). Accordingly, the use of NK-1R antagonists and radiation therapy or chemotherapy, in addition to exerting synergistic antitumor action, decreased the side-effects promoted by both therapeutic strategies [113,114,121]. It has been reported (in in vitro and in vivo experiments) that aprepitant sensitizes acute myeloid cells to the cytotoxic effects of cytosine arabinoside, and hence, the NK-1R antagonist increases the efficacy of the chemotherapeutic drug and decreases its toxicity [123]. In clinical practice, the administration of aprepitant could allow the administration of a low dose of cytosine arabinoside, avoiding its toxicity.

In rodents, carcinogenicity studies have been performed for aprepitant/fosaprepitant [2]. Aprepitant induced skin fibrosarcomas (125–500 mg/kg/day), hepatocellular adenomas/carcinomas (1000–2000 mg/kg/day), thyroid parafollicular cell carcinoma (1000 mg/kg twice daily) and thyroid follicular cell adenomas/carcinomas (5–1000 mg/kg twice daily). It has been suggested that these carcinogenic effects mediated by aprepitant are related with hepatic CYP metabolism. However, as previously reported [124], for the possible treatment of patients with cancer, it is important to note that the dose of aprepitant (extrapolating the aprepitant concentration used in preclinical studies) would be very low (>20 mg/kg/day) compared to the above carcinogenic doses [124]. In addition, it has been suggested that to achieve antitumor action, aprepitant must be administered daily for an extended period of time (for at least 45 days) according to the response to the treatment [124]. In this sense, high doses of aprepitant have been safe and well tolerated (375 mg/day for two weeks; 80 mg/day initially followed by an increase to 125 mg/every third day for seven months; 160 mg/day, for 45 days; 300 mg/day, for 45 days) [85,117,118,125,126]. Consequently, it is crucial to know whether the period of time and suggested dose are efficient and safe in patients suffering from cancer.

In summary, the data show the safety of aprepitant and the specificity of its antitumor action, which also means that due to this specific action, aprepitant could attenuate the well-known non-desirable effects exerted by radiotherapy and chemotherapeutic agents.

## 6. Aprepitant Can Prevent Metastasis after Surgery

For the treatment of cancer (solid tumors), surgical resection is the best option, but when this is believed to be successful, recurrence appears in up to a third of patients and carries a high risk of mortality [127]. In numerous tumor types, surgery promotes locoregional recurrence and postoperative metastases [128]. Moreover, for tumor diagnosis, a biopsy is required, and this procedure also induces, albeit on a smaller scale than surgical resection, neurogenic inflammation and pain (in both mechanisms, SP is involved), which could promote the growth and metastatic spread of cancer cells (by mechanisms mediated by the SP/NK-1R system, since the latter cells overexpress the NK-1R, and in addition, the level of SP is increased in cancer patients) [64,73]. Thus, it is crucial to find strategies aimed at reducing cancer recurrence and metastases after surgery. One of these strategies could be the use of NK-1R antagonists (e.g., aprepitant). In this sense, it is important to note that surgery promotes pain and neurogenic inflammation, and it is known that SP plays an important role in both mechanisms; for example, the peptide promotes the synthesis of inflammatory cytokines by immune cells and stimulates the phosphorylation of NF-κB, which controls the synthesis of pro-inflammatory mediators [47,129,130]. In addition, SP enhances the expression of factors involved in cellular migration (e.g., matrix metalloproteinases 2 and 9) and promotes plasma membrane blebbing, and hence, the peptide promotes the migration of cancer cells; however, NK-1R antagonists block such migration [53,131,132]. Moreover, it is known that during nociceptive and inflammatory mechanisms, the SP/NK-1R system is up-regulated, and this means that surgery procedures, in which pain and inflammation occur, increase the level of SP; this could favor the migration of cancer cells after the binding of the peptide to the NK-1Rs overexpressed in these cells. In fact, the level of SP was increased in patients that suffered tissue injury [133]. Thus, in cancer patients, a high level of SP (in the plasma and tumor microenvironment) can favor the mitogenesis/migration of the remaining tumor cells after surgery, increasing the risk of recurrence and metastasis (Figure 1). The suggested mechanism can explain why, despite an apparently complete resection of the tumor, recurrence and metastasis appear. Accordingly, the administration of aprepitant prior to performing a biopsy (375–750 mg) or prior to and after (375–750 mg in both cases) oncological surgery procedures could decrease inflammation and cancer recurrence and metastasis (including brain metastasis) by the blockade of the NK-1R. This must be demonstrated and confirmed in future studies.

## 7. Aprepitant Can Prevent Thrombosis and Thromboembolism in Cancer Patients

Thromboembolism frequently appears as a cancer-treatment consequence, and cancer-associated thrombosis is related with worse outcomes for patients suffering from cancer (including an increased mortality prognosis); in fact, in these patients, it is the second cause of death [134]. Tumor factors (e.g., duration of cancer, stage, tumor histology and location), treatment-related factors and patient characteristics are some risk factors related to the appearance of thromboembolism [135]. It is important to note that the tumor mass, acting as an endocrine organ, can release SP into the blood, and in fact, the level of SP in the plasma of cancer patients is higher compared to that found in healthy individuals [53,64,73]. Moreover, it is known that platelets express the NK-1R and that SP, via this receptor, promotes degranulation and Ca^++^ intracellular mobilization and mediates platelet aggregation (thrombus formation) [136]; however, NK-1R antagonists (e.g., L-733,060) reduce this aggregation [137]. Thus, according to the previous data, in the future, the use of aprepitant to prevent the formation of the thrombus must be tested and confirmed in patients with cancer.

## 8. Aprepitant in Cancer and Depression

Psychological factors, exposure to diverse stressors, and lifestyle are involved in the development/progression of cancer [138,139]; cancer and depression frequently co-occur, and cancer-risk development has been associated with chronic severe depression. Moreover, depression predicts cancer progression/mortality [140], whereas psychosocial support can decrease depression and may increase the survival time of cancer patients. Acting on the immune system, the mechanisms involved in depression (e.g., a high release of pro-inflammatory molecules) have been suggested to influence cancer progression (e.g., the interleukin-6 plasma level is associated with several types of cancer) [140,141]. In clinical practice, selective serotonin reuptake inhibitors are currently administered for the treatment of patients with cancer-induced depression [140]; however, the use of specific drugs against both depression and cancer is required. These drugs could be NK-1R antagonists (e.g., aprepitant) because it is known that SP, via the NK-1R, promoted the release of pro-inflammatory cytokines; that in depressive patients, a high level of SP was reported in the plasma and tissue; that SP responded acutely to psychological stress in humans; that antidepressant drugs diminished the level of SP in the central nervous system; and that psychotropic drugs altered the expression of genes encoding the synthesis of tachykinins and the NK-1R [47,117,129,130,142,143,144,145]. Thus, the high level of SP in depressive patients with cancer could accelerate the development of the tumor mass, as cancer cells overexpress the NK-1R and the peptide promotes the proliferation of tumor cells. Moreover, it is known that an adequate dose of aprepitant (300 mg/day) can induce an antidepressant effect, and when the dose was increased to 375 mg/day, a decrease in the level of SP was reported [126]. This means that aprepitant, in addition to its antitumor action, decreases inflammation and the level of SP (which is high in depressive patients) and that the SP/NK-1R system is an important link between depression and cancer.

## 9. Conclusions

The SP stimulus is crucial for the survival of tumor cells, and the overexpression of the NK-1R in cancer cells provides an excellent tool for its therapeutic use. Thus, due to this overexpression, NK-1R ligands (e.g., ^213^Bi-DOTA-[Thi^8^,Met(O2)^11^] substance P and radioconjugates of aprepitant) have been developed and used in targeted radionuclide tumor therapy with promising results for selective irradiation to kill cancer cells [6,7,146]. DOTA-SP showed a lower uptake by normal cells than by tumor cells, and this has been related to the higher number of NK-1Rs expressed in the latter cells. Accordingly, we propose the use of one of these NK-1R ligands, the NK-1R antagonist aprepitant, which promotes the death of cancer cells by apoptosis and blocks the migration of these cells. Many experimental data have demonstrated that aprepitant exerts effective antitumor action and that it could be administered to treat tumors independently of both the clinical stage and tumor biology. These data support the reprofiling of aprepitant for a new therapeutic use as an antitumor agent because, in addition, the pharmacokinetics, contraindications, metabolism and safety of this antiemetic drug are well known. To exert an antitumor effect, the dose (>20 mg/kg/day) and days (for a long period of time, according to the response to treatment) of administration of aprepitant must be increased with respect to the current therapeutic antiemetic application (three days: 120 mg, 80 mg, 80 mg) [124]. Dose escalation, tolerability and days of administration must be confirmed in future phase 1 clinical trials, as must the use of aprepitant alone or in combination with chemotherapeutic drugs to treat tumors. As new therapeutic strategies are urgently required for oncological diseases, aprepitant repositioning is a promising and excellent therapeutic alternative for the treatment of any type of cancer. However, as mentioned in a recent review on drug repositioning in oncology [32], regulatory trouble and financial support from governments and funding agencies must be resolved beforehand to stimulate the active participation of the pharmaceutical industry.

The surgical removal of a solid tumor offers the best prospect for a good prognosis, although later metastasis frequently appears and the risk of mortality increases. It is important to note that surgery promotes mechanisms mediated by SP such as the inflammatory response and pain. In other words, the surgical procedure increases the release of SP, which, through the NK-1R, could promote after surgery the growth, migration, spreading and metastasis of the remaining local or distant cancer cells (in which the SP/NK-1R system is up-regulated), and as aprepitant blocks these mechanisms, we suggest the blockade of NK-1Rs (before and after the surgical procedure) with the NK-1R antagonist aprepitant to prevent recurrence and metastasis. Moreover, it is known that paravertebral anesthesia decreased tumor recurrence and metastases, and it seems that this decrease was due to the blockade of the release of SP. Consequently, it is necessary to develop a clinical trial to test the antimetastatic action of aprepitant administered before and after a surgical procedure in patients with cancer. In this way, during the surgical procedure, a double antimetastatic strategy can be suggested to ameliorate the post-operative survival of patients suffering from cancer: the decrease in the release of SP (using paravertebral anesthesia) and the blockade of the NK-1Rs (before and after the surgical procedure) by the NK-1 receptor antagonist aprepitant. Finally, many experimental data suggest the use of aprepitant to prevent thrombosis/thromboembolism in cancer patients, that a link between depression and cancer could occur through the SP/NK-1R system and that this system can be used as a predictive factor in cancer; however, these points must be confirmed in future studies.

This review shows the large number of pre-clinical and clinical data demonstrating the involvement of the SP/NK-1R system in cancer and the potential use of aprepitant as a broad-spectrum antitumor drug. In addition, patents using NK-1R antagonists for the treatment of cancer have been published [29]. It seems that aprepitant could be an intelligent bullet against cancer; however, it is surprising and incomprehensible that pharmaceutical companies have had no interest to date in the SP/NK-1R system as a potential strategy for the treatment of cancer when other research lines have been actively developed.

## Figures and Tables

**Figure 1 cancers-12-02682-f001:**
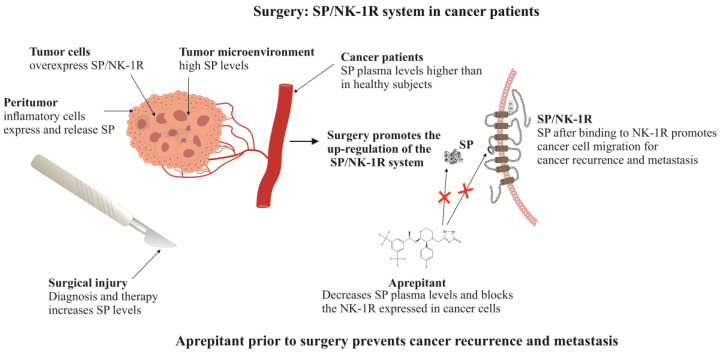
Surgery in cancer patients and the use of aprepitant to prevent cancer recurrence and metastasis.

**Table 1 cancers-12-02682-t001:** SP/NK-1R system as a predictive factor in cancer.

SP/NK-1R expression: used as a predictor for the prognosis of colorectal cancer [75]
SP/NK-1R expression is positively correlated with a higher expression of Ki-67, which is positively correlated with the rate of odontogenic tumor growth [81]
SP expression increased as the proliferation of cancer cells increased [77]
Number of SP positive fibers: correlated with lymph node metastasis, tumor size and cancer differentiation [58,76]
In serum: SP level was higher in cancer patients than in healthy subjects [64,73]
NK-1R: a higher number is related with cancer stage, larger tumor size, tumor-node metastasis, higher invasion/metastatic potential and poor prognosis [40,65,74]
NK-1R (full-length isoform): its expression is inversely correlated with lymph-node metastasis and tumor-lymph-node metastasis [65]
NK-1R (truncated isoform): a high expression is correlated with a poor prognosis [71]
Non-tumor epithelium: SP nuclear/cytoplasmic expression is associated with the presence of epithelial dysplasia and carcinoma [79]

**Table 3 cancers-12-02682-t003:** Safety of aprepitant.

Shows few contraindications (co-administration of cisapride/pimozide, hypersensitivity) [13,111,115]
A relative contraindication occurs when co-administered with dexamethasone, oral hormonal contraceptives, benzodiazepines, ketoconazole or warfarin [111]
Aprepitant’s therapeutic index is very high, and cases showing a toxic effect due to an overdose of the drug are rare [111]
Does not affect QTc intervals [111]
A high dose of aprepitant: proliferation of normal cells was not affected [28,82]
A high dose of aprepitant (300 mg/day; 1140 mg/day for 45 days): safe and well tolerated [5,111,117]
The damage exerted by aprepitant was higher in cancer cells than in normal cells [82]
Lymphocytes: IC_50_ ten-fold higher than that for cancer cells [82]
Fibroblasts: IC_50_ higher than the IC_50_ for cancer cells [83]
Human breast epithelial cells: IC_50_ > 90 μM, three times higher than the IC_50_ for breast cancer cells and higher than the IC_100_ for these tumor cells [17]
Non-tumor cells: IC_50_ (60 μM) and IC_100_ (90 μM); IC_100_ for cancer cells: 50 µM [6,83]

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
