# Peer review of "The Neurokinin-1 Receptor Antagonist Aprepitant: An Intelligent Bullet against Cancer?"

_cancers, 2020, doi:10.3390/cancers12092682_

Round 1
Reviewer 1 Report
A paragraph separating previous experiments in support of aprepitant for anticancer drug. These should include the following:
1. Basic science discovery of the compound and its anti-neurokinin-1 activities.
2. The description of the biology of the anti NK1 antagonist and their various forms.
3. In vitro experiments and description of the various cell lines used to study this.
4. In vitro experiments using various cell systems.
5. In vivo animal experiments.
6. In vivo human preclinical studies.
7. Studies specifically related to cancer.
8. Cell lines in vitro and in vivo models.
9. Human models and human case studies in a tabular format.
Author Response
A paragraph separating previous experiments in support of aprepitant for anticancer drug. These should include the following:
1. Basic science discovery of the compound and its anti-neurokinin-1 activities.
2. The description of the biology of the anti NK1 antagonist and their various forms.
3. In vitro experiments and description of the various cell lines used to study this.
4. In vitro experiments using various cell systems.
5. In vivo animal experiments.
6. In vivo human preclinical studies.
7. Studies specifically related to cancer.
8. Cell lines in vitro and in vivo models.
9. Human models and human case studies in a tabular format.
This has been done (in the new version, changes appear in red). See 2, lines 44-51, 58-81 and 87-90.

Reviewer 2 Report
The main aim of the work “The Neurokinin-1 Receptor Antagonist Aprepitant: an Intelligent Bullet against Cancer?” was to review many anti-cancer effects of aprepitant application. The presented manuscript contains 143 references, 3 tables and one figure. The article is a source of extensive knowledge on the biological behavior of aprepitant (administered alone or in combination regimen with other drugs) in living organisms, both against healthy and neoplastic cells. It gives a broad-spectrum of antitumor aprepitant action. According to the Authors, the presented data (the broad-spectrum antitumor action) support the re-profiling of aprepitant for a new therapeutic use as an antitumor agent. Based on the literature reports discussed in the paper, I consider such a conclusion to be justified. The work is well designed and legibly written. However, I would like to highlight a few shortcomings, one of which (No. (1)) I consider as a very important.
- There is practically nothing in the manuscript about research on the use of aprepitant in nuclear medicine. There is only one mention (in Conclusion) of the 213Bi-DOTA-[Thi8,Met(O2)11] radiobioconjugate studies, but no literature reference is even given! ( Cordier, D.; Forrer, F.; Bruchertseifer, F.; Morgenstern, A.; Apostolidis, C.; Good, S.; Müller-Brand, J.; Mäcke, H.; Reubi, J.C.; Merlo, A. Targeted alpha-radionuclide therapy of functionally critically located gliomas with 213Bi-DOTA-[Thi8,Met(O2)11]-substance P: A pilot trial. Eur. J. Nucl. Med. Mol. Imaging 2010, 37, 1335–1344.). It is also worth paying attention to the newer works of this group on the use of Ac-225 instead of Bi-213. If the submitted review article is to provide an exhaustive answer to the question: Aprepitant: an Intelligent Bullet against Cancer?, the information on the potential use of aprepitant in nuclear medicine cannot be missing!!! There are many articles in the literature on the labeling of non-peptide NK-1R antagonists (including aprepitant) with diagnostic or therapeutic radionuclides (e.g. Pharmaceutics 11, 443 (2019), “The significance of NK1 receptor ligands and their application in targeted radionuclide tumour therapy”; Molecules 25, 3756 (2020), ”Radiochemical synthesis and evaluation of novel radioconjugates of neurokinin 1 receptor antagonist aprepitant dedicated for NK1R positive tumours”). In the former (Pharmaceutics), in the section 3.3. Antagonist Radioligands of NK1R for Targeted Radionuclide Imagin there is a comprehensive overview of NK-1R non-peptide radio-antagonists (including radio-ligand based on aprepitant). The latter (Molecules, published this year) concerns the modification of apepitant molecule and labeling it with a diagnostic (Ga-68) or therapeutic (Lu-177) radionuclide.
I recommend introducing a small section into the manuscript (instead of a small mention in the Conclusions) on the potential use of aprepitant in nuclear medicine and citing at least these three works mentioned above.
- All three tables in the article do not fulfill their role sufficiently. Adding a second column with references to the tables will definitely make it easier for the reader to use the information contained in the tables;
- Line 506: please pay attention to subscripts and superscripts: 213Bi-DOTA-[Thi8,Met(O2)11] substance P instead of 213Bi-DOTA-[Thi8,Met(O2)11] substance P;
- A list of abbreviations would also be helpful.
Summarize, in my opinion, this manuscript can be accepted for publication after taking into account the changes proposed above.
Author Response
REWIEWER 2
There is practically nothing in the manuscript about research on the use of aprepitant in nuclear medicine. There is only one mention (in Conclusion) of the 213Bi-DOTA-[Thi8,Met(O2)11] radiobioconjugate studies, but no literature reference is even given! ( Cordier, D.; Forrer, F.; Bruchertseifer, F.; Morgenstern, A.; Apostolidis, C.; Good, S.; Müller-Brand, J.; Mäcke, H.; Reubi, J.C.; Merlo, A. Targeted alpha-radionuclide therapy of functionally critically located gliomas with 213Bi-DOTA-[Thi8,Met(O2)11]-substance P: A pilot trial. Eur. J. Nucl. Med. Mol. Imaging 2010, 37, 1335–1344.). It is also worth paying attention to the newer works of this group on the use of Ac-225 instead of Bi-213. If the submitted review article is to provide an exhaustive answer to the question: Aprepitant: an Intelligent Bullet against Cancer?, the information on the potential use of aprepitant in nuclear medicine cannot be missing!!! There are many articles in the literature on the labeling of non-peptide NK-1R antagonists (including aprepitant) with diagnostic or therapeutic radionuclides (e.g. Pharmaceutics 11, 443 (2019), “The significance of NK1 receptor ligands and their application in targeted radionuclide tumour therapy”; Molecules 25, 3756 (2020), ”Radiochemical synthesis and evaluation of novel radioconjugates of neurokinin 1 receptor antagonist aprepitant dedicated for NK1R positive tumours”). In the former (Pharmaceutics), in the section 3.3. Antagonist Radioligands of NK1R for Targeted Radionuclide Imagin there is a comprehensive overview of NK-1R non-peptide radio-antagonists (including radio-ligand based on aprepitant). The latter (Molecules, published this year) concerns the modification of apepitant molecule and labeling it with a diagnostic (Ga-68) or therapeutic (Lu-177) radionuclide. I recommend introducing a small section into the manuscript (instead of a small mention in the Conclusions) on the potential use of aprepitant in nuclear medicine and citing at least these three works mentioned above.
This has been done. The three mentioned references (6, 7 and 146) have been added. See pages 2 (lines 74-81), 13 (lines 549-552), 15 (lines 619-624) and 22 (lines 1031-1034). References have been renumbered. In the new version, changes appear in red.
All three tables in the article do not fulfill their role sufficiently. Adding a second column with references to the tables will definitely make it easier for the reader to use the information contained in the tables;
References have been added. See Tables 1 (page 6), 2 (page 8) and 3 (page 10).
Line 506: please pay attention to subscripts and superscripts: 213Bi-DOTA-[Thi8,Met(O2)11] substance P instead of 213Bi-DOTA-[Thi8,Met(O2)11] substance P;
This has corrected. See page 13, line 550.
A list of abbreviations would also be helpful.
This has been corrected. The meaning of the abbreviations used has been indicated along the text when they appeared for the first time. See lines 141, 145-147, 159-162, 186, 234…

Round 2
Reviewer 1 Report
accept this article for cancers- revision improved
Reviewer 2 Report
Please put an abbreviation (5-HT3) after the words: '5-hydroxytryptamine type 3' in line 94